# Spatiotemporal Characteristics and Influencing Factors of Tourism–Urbanization–Technology–Ecological Environment on the Yunnan–Guizhou–Sichuan Region: An Uncoordinated Coupling Perspective

**DOI:** 10.3390/ijerph19148885

**Published:** 2022-07-21

**Authors:** Guangming Yang, Guofang Gong, Yao Luo, Yunrui Yang, Qingqing Gui

**Affiliations:** 1School of Management, Chongqing University of Technology, Chongqing 400054, China; alexandrakung@163.com (G.G.); 18983647867@163.com (Y.L.); yangyunrui@2020.cqut.edu.cn (Y.Y.); 17355937087@163.com (Q.G.); 2Rural Revitalization and Regional High-Quality Development Research Center, Chongqing University of Technology, Chongqing 400054, China

**Keywords:** uncoordinated coupling, spatiotemporal characteristics, influencing factors, Yunnan–Guizhou–Sichuan region

## Abstract

The tourism, urbanization, technology, and the ecological environment both promote and restrict each other. Coordinating the relationship between the four is of great significance to the realization of high-quality sustainable regional development. Taking the Yunnan–Guizhou–Sichuan region as an example, this paper constructs an uncoordinated coupling model for the tourism–urbanization–technology–ecological environment system. Using exploratory spatial analysis and geographic information systems, this paper reveals the temporal and spatial evolution law affecting the uncoordinated coupling relationship between tourism, urbanization, technology and the ecological environment in the Yunnan–Guizhou–Sichuan region from 2010 to 2020, before establishing a panel Tobit model that is used to explore the factors affecting the four systems. The research shows the following: (1) The level of comprehensive development for tourism, urbanization, technology, and the ecological environment in Yunnan, Guizhou, and Sichuan has increased rapidly. Of all these, the tourism industry was the most affected by COVID-19 in 2020, while the level of urbanization, technology, and ecological environment developments in the three provinces has become similar over time. (2) Uncoordinated development between cities is a prominent problem; while the uncoordinated coupling spatial agglomeration in various regions is relatively stable, the proportion of cities with no significant agglomeration form amounts to more than 70%, with mostly low–low (L–L) and high–high (H–H) agglomeration types. (3) The degree to which uncoordinated coupling exists among the four systems in the Yunnan–Guizhou–Sichuan region is affected by many factors. Only eco-environmental pressure has a significant positive correlation with the degree of uncoordinated coupling, while the tourism scale, economic urbanization, eco-environmental response, and investment in technology have a significant negative correlation. These results provide a theoretical basis and practical references for strengthening the government’s macro-control and promoting collaborative regional development.

## 1. Introduction

China’s economy has entered a high-quality development stage; the basis of high-quality development fundamentally lies in economic innovation and green sustainable developments [1]. The tourism industry, urbanization and scientific and technological investments are the driving forces behind economic transformation and development, while the ecological environment forms a basis for ensuring high-quality economic development. Specifically, as a green and efficient industry, tourism plays an important role in high-quality economic development, becoming a new engine with which local governments can promote urbanization. At the same time, it can provide funds for scientific and technological innovation and can also promote the transformation of scientific and technological achievements into productivity [2]. Cities are both important tourist destinations and productive areas of technology research and development. High levels of urbanization and urban economic prosperity have injected vitality into the development of both tourism and science and technology [3]. Technology is the primary productive force; it is a strong driving force behind urban construction and promotes the transformation and upgrading of the tourism industry [4]. In the process of development, the three factors listed above are inseparable from the ecological environment, and a good ecological environment provides basic support for their development [5]. Tourism, urbanization, technology, and the ecological environment both promote and restrict each other. Coordinating the relationship between these four elements is of great significance to the realization of high-quality sustainable regional development.

The term “coupling coordination relationship” refers to the mutual interaction between and influence affecting two or more systems [6]. In the rapid development of an economy, the interaction between tourism development, urban construction, scientific and technological innovation, and the ecological environment presents a non-benign interaction phenomenon that belongs to the stage problem of regional development and which has a certain inevitability. Therefore, this paper identifies the uncoordinated coupling relationship between tourism, urbanization, technology, and the ecological environment; it does so using reverse thinking, constructing a bridge from uncoordinated coupling to coordinated coupling in theory and enriching the theoretical system of coupling and coordination.

The Yunnan–Guizhou–Sichuan region is rich in tourism resources; it is also an important area of China’s western development and the Yangtze River Economic Belt; however, the differences in the geographical location, resource endowment and economic levels have led to an imbalance in the whole region’s development process, which seriously restricts the coordinated development of tourism, urbanization, technology, and the ecological environment in the region. In view of this, this study takes the Yunnan–Guizhou–Sichuan region as its research area, constructing an uncoordinated coupling model of the “tourism–urbanization–technology–ecological environment”; it analyzes the uncoordinated coupling relationship and main influencing factors and provides suggestions and countermeasures for a better realization of the coordinated development of the tourism–urbanization–technology–ecological environment in the Yunnan–Guizhou–Sichuan region.

## 2. Theoretical Background

### 2.1. Literature Review

Research on the interactive relationship between tourism, urbanization, technology and the ecological environment is in its early stages and has produced valuable research results; however, most studies have predominantly focused on pairwise systems—there are relatively few studies on the relationship between the four systems. The earliest research on the relationship between tourism and the environment began in the 1920s; of this, Meinecke’s [7] research on the environmental impact of tourism activities in California’s Sequoia National Park is the most enlightening. Stansfield [8] then discussed the importance of urban tourism research from a new perspective, giving people a new understanding of the relationship between the development of tourism and urbanization. Gladstone [9] analyzed some characteristics of tourism urbanization in the United States and metropolitan leisure areas. Qian, et al. [10] believed that the development of tourism helps to improve the level of urbanization, while Li, et al. [11] by studying the coupling and coordination relationship between tourism, urbanization and ecological environment in Chongqing from 2000 to 2017, concluded that the interaction between tourism development and urbanization is significant.

Some Western countries that had experienced earlier industrial development took the lead in studying the relationship between urbanization and the ecological environment. Howard [12] first proposed and outlined the garden city theory, while Shahbaz, et al. [13] studied the relationship between urbanization and energy consumption in Malaysia in depth. Huang, et al. [14] analyzed the coupling coordination degree between urbanization and the ecological environment in Kazakhstan using the coupling coordination degree model; they found that the coupling coordination degree between the two systems shows an upward trend. Yu [15] took China as an example, conducting an empirical test of urbanization and the ecological environment; his research showed that there is an inverted “U” shape between the two systems.

Poon’s [16] research showed that information technology does not affect tourism itself (rather, it affects its organization and management), and that it is widely used in tourism. Stamboulis and Skayannis [17] believed that tourism is undergoing profound changes and facing various challenges. To accelerate tourism’s sustainable development, we should pay attention to innovative tourism technology. Cosma, et al. [18] took rural Romania as an example, analyzing how rural tourism companies drive service innovation, management innovation and product innovation through technological innovation. Rodríguez, et al. [19] observed that one of the main driving forces affecting the development of tourism is information technology innovation, stating that the driving effect of information technology on tourism will become more and more significant. Rafael [20] introduced the trends in and dynamic conclusions of a smart tourism and smart destination survey, analyzing the impact of technology and sustainability on the tourism industry.

When studying the environmental impact of ecotourism, Wackernagel and Yount [21] took the lead in applying the ecological footprint theory with the purpose of evaluating sustainable tourism development. Hunter [22] clearly put forward the concept of tourism’s ecological footprint, while Ozturk, et al. [23] used the environmental Kuznets curve hypothesis to explore the interaction between tourism’s development and its ecological footprint. Li and Lv [24] used panel data for 95 countries from 2000 to 2014 to investigate the impact of tourism development on carbon dioxide emissions. The results confirmed that the development of tourism has a significantly enhanced impact on carbon dioxide emissions.

### 2.2. The Uncoordinated Coupling Mechanism

Coupling refers to the phenomenon whereby two (or more) systems or elements affect each other through their interaction [25]. In an analysis, the coupling degree is used to describe the degree of interaction between the systems; in this way, the relationship between systems or elements can be quantified. Coordination is the interactive relationship between effective circulation, promotion between two (or more) systems or system elements. The degree of coordination and coupling is used to measure the degree of benign interaction between systems or elements, reflecting the change trend from disorder to order [26]. The uncoordinated coupling degree is a problematic state in the interaction relationship; it shows the mutual inhibition between systems or elements.

Tourism, urbanization, technology, and the ecological environment form a complex system composed of their respective subsystems, which interact and blend with each other (Figure 1). A thorough understanding of the relationship between and the interaction mechanism affecting the four systems is of great significance to the study of the uncoordinated coupling of tourism, urbanization, technology, and the ecological environment.

As shown in Figure 1, the tourism subsystem brings development impetus to urbanization through industrial development, gathers a population and provides more employment opportunities; however, it also hinders the further promotion of urbanization due to its large consumption of resources and occupation of public space. In turn, the urbanization subsystem provides various support conditions for tourism development; however, it can also threaten the development of tourism through excessive commercialization and other behaviors [27].

The emergence of new forms of tourism forces enterprises to pay more attention to R&D and to the application of innovative scientific and technological products; it also causes them to focus on the development of new products and services and improve the level of tourism services offered. Scientific and technological innovations can continuously improve the quality of tourism products and realize both industrial integration and smart tourism; however, technology may also cause damage to both the natural and the human tourist destination environment, while the sustainable development of tourist destinations becomes difficult.

Tourism is closely related to the ecological environment, and relying on the natural landscape and a high-quality environment can help development; however, when tourism develops rapidly, resulting in over-exploitation and more and more tourists, it is easy to exceed the ecological environment’s carrying capacity and forget to limit the development of tourism; however, new tourism methods (such as low-carbon tourism and ecotourism) can effectively publicize environmental awareness and feed back into the ecological environment [28].

The development of urbanization has effectively gathered talented individuals engaged in scientific and technological innovation, as well as providing suitable scientific research conditions and the necessary financial support. Scientific and technological innovation promotes the upgrading of the industrial structure, encourages the establishment of smart cities and drives the process of urbanization by improving labor productivity and creating demand; however, urbanization, which naturally occurs in line with the development of technology, also brings many problems, such as traffic congestion, decline in the quality of housing and human settlements, the occupation of agricultural space, and urban safety risks [29].

The ecological environment subsystem provides perfect basic conditions for scientific and technological innovation, which protects the ecological environment by optimizing industrial structures and improving the efficiency of resource utilization; however, technology also give birth to new material needs, resulting in the waste of natural resources and destruction of the natural environment, increasing the pressure bearing on and polluting the environment.

The ecological environment subsystem provides the necessary supporting conditions for the development of urbanization. In the process of development, the urbanization subsystem will inevitably destroy the ecological environment, over-consume resources, occupy ecological space, and cause the excessive discharge of pollutants, etc., all of which will seriously damage the quality of the ecological environment. Once the ecological environment is irreparable or difficult to repair, the process of urbanization will be limited, forcing the exploration of the path of transformation and upgrades. Currently, urbanization provides sufficient capital, technology, and policy support for ecological governance [30].

It can be seen that the four subsystems are closely linked and interact with each other; however, there will inevitably be periodic friction and imbalance between their development. Ultimately, they will need to rely on mutual cooperation and common improvement if they are to move from disorder to order and promote the sustainable and stable development of tourism, urbanization, technology, and the ecological environment.

## 3. Study Area

The Yunnan–Guizhou–Sichuan region is located in the southwest of China and has a complex terrain; it is adjacent to the Guangxi Zhuang Autonomous Region and Guizhou province to the east, Sichuan province across the river to the north, the Tibet Autonomous Region to the northwest, Myanmar to the west, and Vietnam and Laos to the southeast and south, respectively (Figure 2). At the same time, the Yunnan–Guizhou–Sichuan region is also the key construction area of the country’s western development strategy. Due to its special geographical location, rich natural resources and the support of national policies, we have chosen this as our research object.

In terms of tourism, the Yunnan–Guizhou–Sichuan region is rich in a large number of cultural and natural tourism resources (such as the ancient cities of Lijiang and Langzhong, located in Yunnan province and Sichuan province, respectively). Guizhou province is famous across the world for its unique karst landforms. By the end of 2020, the total tourism revenue for the Yunnan–Guizhou–Sichuan region was 1943 billion yuan, accounting for 87% of the country’s total domestic tourism revenue. A total of 1.599 billion tourists were received, accounting for 55.3% of the total number of tourists in China. The region contains 1610 Class A scenic spots, accounting for 12.1% of such spots nationally, and houses 3154 travel agencies (7.8% of the national total).

In terms of urbanization, the gross domestic product (GDP) of the three provinces is 9094 billion yuan in 2020, accounting for 9% of GDP. The urbanization rate in Sichuan province reached 56.73% in 2020, higher than that of Yunnan and Guizhou province, but far lower than the national urbanization rate of 63.89%. The urbanization construction in the Yunnan–Guizhou–Sichuan region still has great room to make progress.

From a technology perspective, 27,225 technology contracts have been signed within the three provinces in 2020, accounting for less than 5% of national technology contracts. These consist of 20,456 technology contracts in Sichuan province and less than 4000 in Yunnan and Guizhou provinces. The three provinces’ R&D expenditure amounts to 146.3 billion yuan, accounting for less than 6% of domestic R&D expenditure. Sichuan province’s R&D expenditure is more than three times that of Yunnan and Guizhou provinces; it can therefore be seen that the scientific and technological development level of the Yunnan, Guizhou and Sichuan provinces is low.

In terms of the ecological environment, in 2020 the national forest coverage rate was only 23.04%, while the forest coverage rate within Sichuan province was 40.03%. The forest coverage rate within Yunnan province and Guizhou province was over twice the national forest coverage rate. There are 62 national nature reserves in the region, accounting for 13.1% of the national total. In 2020, the total afforestation area in China was 6.77 million hectares, while the total afforestation area in the three provinces of Yunnan, Guizhou and Sichuan was 103.26, accounting for 15.3% of such areas nationwide. Overall, the ecological environment within the Yunnan, Guizhou and Sichuan provinces is good.

## 4. Methods

### 4.1. Methodology

#### 4.1.1. Evaluation Model for the Comprehensive Development Level

This paper adopts the entropy method [31] to objectively weight the system evaluation index. The entropy method has the following advantages in determining the index weight: First, the result is intuitive and easy to understand, and the method is practical; Second, the method avoids the interference of subjective factors and has strong objectivity; Third, there is no limit on the number of indicators and the method’s scope of application is wide.

In order to make the comparison between different years more reasonable, the time variable h is added. The specific steps are as follows:

(i)Build the original data matrix: If there are h years, m provinces and N indicators, the matrix of a year a is $X=\left\{x_{ij}\right\}m\times n\;\left(1\leq i\leq m,1\leq j\leq n\right)$. Here, $x_{ij}$ is the index value of the jth index of the ith city.(ii)Since there are differences in the dimension and order of magnitude of each index, it is necessary to use the dimensionless process on each index:(1)$$\mathrm{Positive}\;\mathrm{indicators}:\;Z_{ij}^{\prime }=\frac{x_{ij}-min\left\{x_{ij}\right\}}{max\left\{x_{ij}\right\}-min\left\{x_{ij}\right\}}$$
(2)$$\mathrm{Negative}\;\mathrm{indicators}:\;Z_{ij}^{\prime }=\frac{max\left\{x_{ij}\right\}-x_{ij}}{max\left\{x_{ij}\right\}-min\left\{x_{ij}\right\}}$$

In order to eliminate the influence of 0, add 0.01 to the dimensionless data to obtain the final $Z_{ij}$. Next:

(i)Normalize the indicators:(3)$$P_{ij}=Z_{ij}/\sum_{a=1}^{h}\sum_{i=1}^{m}Z_{aij}$$(ii)Calculate the entropy of each index:(4)$$E_{j}=-k\sum_{a=1}^{h}\sum_{i=1}^{m}P_{aij}lnP_{aij},\;k=1/ln(hm)$$(iii)Calculate the entropy redundancy of each index:(5)$$D_{j}=1-E_{j}$$(iv)Calculate the weight of each index:(6)$$W_{j}=D_{j}/\sum_{j=1}^{n}D_{j}$$

The comprehensive development level of tourism, urbanization, technology, and the ecological environment in the Yunnan–Guizhou–Sichuan region from 2010 to 2020 is calculated by weighting.
(7)$$U_{\left(1,2,3,4\right)}=\sum_{j=1}^{m}w_{j}u_{ij}$$

Here, $U_{1}$, $U_{2}$, $U_{3}$ and $U_{4}$ are the comprehensive evaluation index values for the tourism, urbanization, technology, and ecological environment subsystems, respectively, in Yunnan, Guizhou and Sichuan. Where $w_{j}$ represents the weight of each index in the four systems and $u_{ij}$ is the functional contribution of the index j to the system.

#### 4.1.2. Uncoordinated Coupling

(i)Construction of the uncoordinated coupling model

The uncoordinated coupling model needs to be based on the coordination model. Therefore, this paper first constructs the coupling model, as shown below [32]:(8)$$C={\left\{\frac{U_{1}\times U_{2}\times U_{3}\times U_{4}}{{\left[(U_{1}+U_{2}+U_{3}+U_{4})/4\right]}^{4}}\right\}}^{1/4}$$
(9)$$D=\sqrt{C}\times \sqrt{T}$$
(10)$$T=a\times U_{1}+b\times U_{2}+c\times U_{3}+d\times U_{4}$$
(11)ND=1−D

In the formula, the coupling coordinated development degree is expressed in D, the coupling degree is expressed in C, the uncoordinated coupling degree is expressed by ND, $U_{1}$ represents the comprehensive development level of tourism, $U_{2}$ represents the comprehensive development level of urbanization, $U_{3}$ represents the comprehensive development level of technology, and $U_{4}$ represents the comprehensive development level of the ecological environment; The comprehensive evaluation index of the tourism–urbanization–technology–ecological environment system is expressed by *T*. Here, a, b, c, and d represent undetermined coefficients, because the four systems are equally important in this study. The undetermined coefficients are assigned 0.25.

(ii)Classification standard for the degree of uncoordinated coupling

The degree of uncoordinated coupling is divided into four levels: high-level uncoordinated coupling; running in uncoordinated coupling; antagonistic uncoordinated coupling; and low-level uncoordinated coupling [33]. Table 1 gives the specific scope divisions.

#### 4.1.3. Spatial Autocorrelation Analysis

Exploratory spatial analysis (ESDA) is divided into global and local correlation; it can be used to analyze the spatial agglomeration phenomenon and spatial correlation between various variables and to study the correlation and dependence between data from the location of space [34]. In view of this, Moran’s I index is used to study the spatial correlation characteristics of high and low values in some areas of Yunnan, Guizhou, and Sichuan.

The global Moran’s I index is:(12)$$MI=\frac{n\times \sum_{i=1}^{n}\sum_{j\neq 1}^{n}W_{ij}\left(x_{i}-\bar{x}\right)(x_{j}-\bar{x})}{(\sum_{i=1}^{n}\sum_{j=1}^{n}W_{ij})\times \sum_{i=1}^{n}{\left(x_{i}-\bar{x}\right)}^{2}}$$

In the above formula, MI represents the Moran index; n represents the number of study provinces; $x_{i}\;\mathrm{and}\;x_{j}$ respectively represent the uncoordinated coupling degree at i and j positions of provinces, while $\;\bar{x}$ represents the average value of the uncoordinated coupling degree; and $W_{ij}$ indicates the neighborhood relationship between  i and j in the province. When i and j are adjacent, $W_{ij}=1$, otherwise it is 0. The numerical range of the global Moran index is [−1, 1], where greater than 0 indicates a positive spatial correlation, less than 0 indicates a negative spatial correlation, and equal to 0 indicates it is irrelevant.

Global autocorrelation assumes spatial homogeneity, which cannot reflect the characteristics of local agglomeration; further local spatial autocorrelation analysis is needed. The formula for the local spatial correlation index (LISA) is as follows:(13)$$MI_{i}=\frac{\left(x_{i}-\bar{x}\right)}{m_{0}}\sum_{j}W_{ij}\left(x_{j}-\bar{x}\right)$$

In the above formula, $x_{i}$ denotes the uncoordinated coupling degree of city i; $\bar{x}$ denotes the average value of the uncoordinated coupling degree of cities; $MI_{i}>0$ denotes the spatial clustering of observations similar to the uncoordinated coupling degree of a city (H–H or L–L); and $MI_{i}<0$ denotes the spatial clustering of observations not similar to the uncoordinated coupling degree of a city (L-H or H-L).

#### 4.1.4. The Tobit Model

The Tobit model is divided into the mixed-effect Tobit model, random-effect Tobit model and fixed-effect Tobit model. The Tobit model can effectively solve the problem of the limited value of the dependent variable in non-cooperative coupling. Therefore, in the process of data modeling, the mixed-effect Tobit model, random-effect Tobit model, and fixed-effect Tobit model are used to determine whether the constructed model results are true and reliable. The models are selected through a comparative analysis of the likelihood-ratio test, Hausmann test and F test. The specific models are as follows [35].

(i)The mixed-effect Tobit model


(14)
$$Y_{i}^{*}=\alpha +\beta_{1}X_{1}+\beta_{2}X_{2}+\beta_{3}X_{3}+\beta_{4}X_{4}+\beta_{5}X_{5}+\beta_{6}X_{6}+\beta_{7}X_{7}+\beta_{8}X_{8}+\beta_{9}X_{9}+\beta_{10}X_{10}+\beta_{11}X_{11}+\varepsilon_{it}$$


In Formula (14), $Y_{it}^{*}$ is the latent variable in the model (that is, the development level of uncoordinated coupling); $X_{1},\;X_{2},X_{3},X_{4},X_{5},X_{6},X_{7},X_{8},X_{9},X_{10}\;\mathrm{and}\;X_{11}$ represent the number of domestic tourists, domestic tourism income, number of star hotels, per capita GDP, urbanization rate, investment in real estate development, industrial sulfur dioxide emission, harmless treatment rate of urban domestic waste, forest coverage rate, technology expenditure and the number of college students; $\beta_{1},\beta_{2},\beta_{3},\beta_{4},\beta_{5},\beta_{6},\;\beta_{7},\beta_{8},\beta_{9},\beta_{10}\;\mathrm{and}\;\beta_{11}$ are the coefficients of each independent variable in the model, and their values reflect the internal relationship between the uncoordinated coupling development of various influencing factors; finally, $\varepsilon_{it}$ is the random disturbance term used in the model.

(ii)Tobit random-effects model


(15)
$$Y_{i}^{*}=a_{i}+\beta_{1}X_{1}+\beta_{2}X_{2}+\beta_{3}X_{3}+\beta_{4}X_{4}+\beta_{5}X_{5}+\beta_{6}X_{6}+\beta_{7}X_{7}+\beta_{8}X_{8}+\beta_{9}X_{9}+\beta_{10}X_{10}+\beta_{11}X_{11}+u_{i}+\varepsilon_{it}$$


In Formula (15), $u_{i}\;$refers to the individual effect, which does not change with time and cannot be observed.

(iii)Tobit fixed-effect model
(16)$$Y_{i}^{*}=a_{i}+\beta_{1}X_{1}+\beta_{2}X_{2}+\beta_{3}X_{3}+\beta_{4}X_{4}+\beta_{5}X_{5}+\beta_{6}X_{6}+\beta_{7}X_{7}+\beta_{8}X_{8}+\beta_{9}X_{9}+\beta_{10}X_{10}+\beta_{11}X_{11}+\varepsilon_{it}$$

### 4.2. Indicator Selection and Data Sources

Constructing a scientific and reasonable evaluation index system is the basis of an evaluation of the level of coordinated development. Based on the actual development of the tourism, urbanization, technology, and eco-environment system and referring to the existing literature [11,27,31,36], as well as following the principles of scientificity, representativeness, and availability, the comprehensive evaluation indexes for the four subsystems are constructed. The specific evaluation index system is shown in Table 2.

The data for the indicators in this paper come from the statistical bulletin of national economic and social development, Yunnan statistical yearbook, Sichuan statistical yearbook, Guizhou statistical yearbook, and China urban statistical yearbook from 2010 to 2020. Where data is lacking for some years, the simple moving average method is used to complete them.

## 5. Results

### 5.1. Measurement of the Tourism Development Level

Formula (7) is used to obtain the comprehensive evaluation index value for tourism in Yunnan, Guizhou and Sichuan, as shown in Figure 3.

From 2010 to 2019, the comprehensive evaluation index for the tourism industry in Yunnan, Guizhou and Sichuan showed both an upward and downward trend; overall, the development trend increased; however, in 2020, due to the impact of COVID-19, the development trend for tourism in the three provinces and their cities fell sharply.

Taking prefecture level cities as the research object, ArcGIS 10.3 is used to display the comprehensive development evaluation index for tourism in a spatial distribution map. As shown in Figure 4, tourism development prior to 2016 was mainly concentrated in Chengdu, Kunming, Guiyang, and other economically developed areas or their surrounding cities. Since 2016, tourism in other areas that are rich in tourism resources has increased rapidly. Taking Chengdu as the starting point, Sichuan drives the development of tourism in the surrounding prefectures of Ganzi and Aba, as well as in other areas. In terms of behavioral tourism, tourism development in Guizhou is relatively stable and is mainly concentrated in Sanqian and Guiyang. Tourism development in other areas is relatively low-key. Yunnan has the ancient city of Lijiang, one of the four ancient cities in China; it is rich in tourism resources, which are mainly concentrated in Kunming, Lijiang, Dali, Xishuangbanna, and Diqing. The development of tourism in the border areas of Nujiang and other provinces is relatively poor.

### 5.2. Measurement of the Urbanization Development Level

The comprehensive evaluation idex values of urbanization in Yunnan, Guizhou and Sichuan are obtained from Formula (7), as shown in Figure 5.

From 2010 to 2020, the comprehensive evaluation index of urbanization in Yunnan, Guizhou and Sichuan showed a straight-line growth trend (apart from a decrease in Guizhou province’s trend in 2014). Between 2010 and 2014, the urbanization development index for Guizhou province was the highest. During this period, Sichuan province’s urbanization index increased rapidly, becoming the same as that of Guizhou in 2014. Following this, Yunnan province’s comprehensive index increased the fastest and most consistently, coming level with that of Sichuan province in 2020.

Taking the urbanization development of the various regions in five different years (2010, 2013, 2016, 2019, and 2020) as an example, this paper introduces the urbanization development of various regions (Figure 6). From the perspective of space, the urbanization development level is uneven between regions, with Chengdu, Kunming, and Guiyang at the center, followed by Zunyi, Luzhou, Nanchong, Qujing, and a few other regions in the second echelon. Guang’an, Liangshan, Pu’er, and Bijie’s urbanization construction falls into the third echelon, while the urbanization construction process for Guangyuan, Bazhong, Diqing, Wenshan and a few other places is at the low-level development stage. These areas are far away from the central cities of economic development, while inconvenient transportation hinders the urbanization development process.

### 5.3. Measurement of the Technology Development Level

Formula (7) is used to obtain the scientific and technological comprehensive evaluation index value for the Yunnan–Guizhou–Sichuan region, as shown in Figure 7.

As shown in Figure 7, the scientific and technological comprehensive evaluation indices for the three provinces show a straight-line upward trend, and the development tracks of the three provinces almost coincide.

As shown in Figure 8, there is an imbalance in the level of technology development among the 46 cities, and overall there was a better development in 2013, and the overall layout of the development over time is more stable. From 2013 onwards, the central cities of Chengdu and Kunming form a technology development circle, mainly including Deyang, Mianyang, Qujing, Yuxi, Zunyi. The development of technology in the border areas of Yunnan, Guizhou and Sichuan provinces is the slowest, as most of these areas are autonomous states, such as Ganzi, Aba and Wenshan. Poor economic development, by geographical location and other restrictions, education, technology resources are insufficient, and the development is more backward. Most of these areas are limited by geographical location, lack of education and scientific and technological resources, and their development is relatively low-level.

### 5.4. Measurement of the Ecological Environment Development Level

Formula (10) is used to obtain the comprehensive evaluation index value of the ecological environment, as shown in Figure 9.

According to the analysis in Figure 10, from 2010 to 2020 the comprehensive evaluation index for the ecological environment in Yunnan, Guizhou, and Sichuan showed an upward and downward fluctuating trend; overall, however, it showed a growth trend. It can be seen that the comprehensive evaluation index for the ecological environment in the three provinces was quite different prior to 2017. Between 2017 and 2020, the development of the comprehensive evaluation index for the ecological environment in the three provinces reached a high level and maintained a steady growth.

The comprehensive level for the ecological environment’s development between 2010 and 2020 is good, with development mainly concentrated in the western and southeast regions (such as Ya’an, Xishuangbanna, southeast Guizhou and a few other regions). Most of these regions are dominated by agriculture and possess valuable tourism resources. Relying on the development of tourism, the comprehensive development level of the ecological environment is high. In addition, Yibin, Zigong, Yuxi, Zunyi, Bijie and a few other regions have prominent problems such as industrial agglomeration, inadequate sewage treatment facilities, and direct sewage discharge; these have damaged the ecological environment and have led to its poor development.

### 5.5. Uncoordinated Coupling Degree Analysis

#### 5.5.1. Temporal Evolution Analysis

According to the uncoordinated coupling model, Formulas (11)–(14) are used to calculate the coupling degree index (C) and the uncoordinated coupling index (ND) for tourism, urbanization, technology and the ecological environment in the Yunnan–Guizhou–Sichuan region, as well as to analyze the uncoordinated coupling characteristics and spatial agglomeration of the time and spatial levels.

To comprehensively analyze the uncoordinated level of the tourism, urbanization, technology, and eco-environmental system in Yunnan, Guizhou, and Sichuan, the uncoordinated coupling level is divided by referring to the classification standard of the uncoordinated coupling level in Table 1 and the uncoordinated coupling measure in Table 3. The results are as follows:

The overall development level of the three provinces of Yunnan, Guizhou, and Sichuan is good, and the uncoordinated coupling degree is in the range of 0.1–0.7. The uncoordinated coupling degree for the tourism, urbanization, technology, and ecological environment system in the three provinces gradually decreases, indicating that the level of uncoordinated coupling among the three provinces is the same, and that the interval development is relatively coordinated.

From 2010 to 2020, the uncoordinated coupling degree for Yunnan and Guizhou provinces experienced three stages: a running in uncoordinated coupling stage; an antagonistic uncoordinated coupling stage; and a low-level uncoordinated development coupling stage. Yunnan province was in the running in uncoordinated coupling stage from 2010 to 2012; in comparison, Guizhou province entered the antagonistic uncoordinated coupling stage in 2012. Both provinces entered the low-level uncoordinated coupling stage in 2018. Sichuan province moved from the running in uncoordinated coupling stage to the antagonistic uncoordinated coupling stage in 2012, then entered the low-level uncoordinated coupling stage in 2018, finally changing to the antagonistic uncoordinated coupling stage in 2020. The detailed analysis is as follows.

According to the calculation results of the above four-subsystem comprehensive evaluation indices, Yunnan Province has the largest difference between the comprehensive evaluation indices for technology and the ecological environment in 2012, and the uncoordinated interaction between them is obvious. In 2018, there is little difference between the comprehensive evaluation indices for tourism and urbanization in Yunnan province. The comprehensive evaluation indices for urbanization and ecological environment are quite different, and the uncoordinated interaction between the two is obvious.

In 2011, the largest difference is found between the comprehensive evaluation indices for tourism and technology in Guizhou province, and the uncoordinated interaction between them is obvious. By 2018, the largest difference in the values of the comprehensive evaluation index of Guizhou province was between tourism and ecological environment.

In 2011, the comprehensive evaluation index values of tourism and technology in Sichuan Province were the biggest difference, which was obviously uncoordinated. In 2018, the reason for the uncoordinated coupling in Sichuan Province was the large difference in the values of the comprehensive evaluation indices of tourism and ecological environment.

#### 5.5.2. Spatial Evolution Analysis

Figure 11 shows that, in comparison to the provincial scale, the uncoordinated coupling degrees for municipal cities in the Yunnan–Guizhou–Sichuan region was high between 2010 and 2020. In 2010, most cities were in the stage of high-level uncoordinated coupling, and only a few cities were running in uncoordinated coupling, indicating that the coordinated development between the tourism, urbanization, technology, and ecological environment systems in most cities was weak at this time. Between 2013 and 2016, the coordinated development of most cities achieved certain results; most cities have moved from high-level uncoordinated coupling to running in uncoordinated coupling. The uncoordinated coupling degrees for Chengdu, Kunming, Guiyang and the surrounding cities were the first to enter the antagonizing uncoordinated coupling stage, driving the coordinated development of the surrounding areas to a certain extent. Between 2019 and 2020 (although Diqing, Nujiang and certain other regions are still in the high-level uncoordinated coupling stage), the level of uncoordinated coupling in Kunming and certain other cities has reached the lowest level in history, entering the low-level uncoordinated coupling stage, with other cities entering the running in uncoordinated coupling stage. Affected by the epidemic in 2020, low-level regions such as Mianyang, Guiyang, and Chengdu have returned from the low-level uncoordinated coupling development level to the antagonistic uncoordinated coupling stage. In short, the interaction between tourism, urbanization, technology, and the ecological environment in various regions has gradually strengthened, but it has not been synchronized within the development process, and the coordinated development between regions needs to be further optimized.

Spatial agglomeration characteristics of uncoordinated coupling: Global autocorrelation

In order to further study the spatial correlation of the uncoordinated coupling adjustment of the tourism, urbanization, technology, and ecological environment system in the Yunnan–Guizhou–Sichuan region, the global Moran’s I index is calculated. The global Moran’s I index is greater than 0, and the standardized statistical indices Z and *p* pass the test, which shows that the Moran’s I index is significant (that is, there is a significant positive spatial autocorrelation in the uncoordinated coupling of tourism, urbanization, technology and ecological environment). In addition, the global Moran’s I index fluctuates up and down, indicating that its spatial agglomeration trend weakens first and then increases. Taking 2013, 2016, and 2019 as the time points, this paper calculates the Moran’s I index for the uncoordinated coupling development of cities in Yunnan, Guizhou, and Sichuan, as shown in Table 4.

The uncoordinated coupling levels for tourism, urbanization, and the scientific and technological transformation of the ecological environment in 46 prefecture-level cities in the three provinces of Yunnan, Guizhou and Sichuan weakens over the period of study, but the spatial agglomeration effect still exists. In subsequent developments, it will be necessary to strengthen the coordinated development between regions.

2.Spatial agglomeration characteristics of uncoordinated coupling: Local autocorrelation

The local spatial correlation index of the uncoordinated coupling degree of tourism, urbanization, technology, and the ecological environment in the Yunnan–Guizhou–Sichuan region from 2010 to 2020 is calculated. The data are spatially visualized using ArcGIS software, and the LISA diagram of the uncoordinated coupling degrees for tourism, urbanization, technology, and the ecological environment in the Yunnan–Guizhou–Sichuan region is obtained, as shown in Figure 12.

From 2010 to 2013 and from 2013 to 2016, the high-concentration areas (H–H-type areas) are mainly concentrated in Sichuan and Yunnan provinces. Generally speaking, they are concentrated in several autonomous regions, which is basically spatially consistent with the economically underdeveloped cities. From 2010 to 2013, there were four cities of this type, then three from 2013 to 2019 and one from 2019 to 2020, with little change seen in quantity. The uncoordinated coupling degrees for these cities are also higher than those of other cities. Most of these cities are located in the marginal areas of Yunnan and Sichuan provinces. The traffic is relatively blocked, which makes both the development of tourism and urbanization slow. At the same time, the surrounding areas also belong to the low-development level of tourism and urbanization, so a low-lying gathering area is formed. A low level of economic development, poor scientific and technological strength, a low level of urbanization, a large proportion of land used for agricultural purposes and underdeveloped transportation hinder the development of tourism to a certain extent, which would otherwise usually be the short route to achieving regional coordinated development.

From 2010 to 2016, Chengdu and Kunming, the main cities in the low–high concentration area (L-H-type areas) did not fluctuate greatly in spatial distribution. The uncoordinated coupling degrees for this type of city are at a low level; however, there is a gap between this type and the low-level type, with room to reduce the uncoordinated coupling degree. In addition, this type of city is restricted by the surrounding cities that have a large degree of uncoordinated coupling; however, it disappeared after 2016, which shows that the coordinated development of the region has achieved good results.

The uncoordinated coupling degrees for low-level agglomeration (L–L) cities are far lower than those of their surrounding cities, forming the growth pole of regional coordinated development. From a spatial pattern distribution perspective, the uncoordinated coupling degrees have changed significantly in 2016, mostly concentrated in Guizhou, Guiyang, Southeast Guizhou and certain other regions; this formed a low-value agglomeration area of regional uncoordinated coupling development, and shows that the protection and development level of the Yunnan–Guizhou–Sichuan region is stable, which makes its uncoordinated coupling development low and coordinated coupling strong, forming a regional core. At the same time, the region is close to the provincial capital city, which has a relatively developed economy, close economic ties between cities, and significant spillover effects (such as talent flow and technology diffusion), thus bringing the surrounding areas up.

### 5.6. Analysis of the Influencing Factors

The Stata model is used to empirically analyze the influencing factors of uncoordinated coupling degrees in Yunnan, Guizhou, and Sichuan. In order to ensure the reliability of the model, the mixed-effect, random-effect and fixed-effect Tobit models are constructed and the optimal model is selected; this model is tested using the LR, Hausman, and F tests. The results are shown in Table 5.

The results in Table 5 show that the *p* values of both the F test and the Hausman test are less than 0.01, indicating that a Tobit regression using the fixed-effect model is reliable.

The analysis results from the fixed-effect Tobit model show there is a significant positive correlation between eco-environmental pressure and degrees of uncoordinated coupling, while the tourism scale, economic urbanization, social urbanization, eco-environmental response, and technology investment all have a significant negative correlation with the level of uncoordinated coupling. Although the urbanization and forest coverage rates and scientific and technological output also have a negative correlation with the level of uncoordinated coupling, it is not significant.

The tourism scale has a significant negative correlation at a 1% level of significance. An increase in domestic tourists will reduce the uncoordinated coupling of tourism, urbanization, technology, and the ecological environment. The expansion of the tourism scale represents the continuous improvements to tourism’s development level, the continuous attention paid to the popularity of tourism, the improvement of infrastructure and transportation facilities, the continuous improvement of tourism service quality, and the continuous strengthening of the awareness of ecological environment protection. In effect, this provides more jobs and employment opportunities for local residents, promotes local economic development, and is conducive to tourism and urbanization and the coordinated development of the technology and ecological environments.

Economic urbanization has a significant negative correlation at a 1% level of significance. An increase in regional per capita GDP will lessen the uncoordinated coupling of tourism, urbanization, technology, and the ecological environment. The reason for this is that the increase in per capita GDP indicates that there has been an increase in residents’ income. People’s concepts of living and consumption are constantly changing. They are no longer satisfied only by the fulfillment of material needs, and so begin to pursue spiritual needs; this makes the tourism industry develop rapidly, results in diversified development to meet people’s differential needs, greatly promotes the healthy development of the tourism market and encourages the coordinated development of various regions.

Social urbanization has a significant negative correlation at a 1% significance level; this means that an increase in real estate investment will reduce the uncoordinated coupling degrees for tourism, urbanization, technology, and the ecological environment. Investment in real estate development shows that an increase in the resident and urban population can create a strong engine for regional development, provide a labor force for urbanization purposes, result in the discovery of scientific and technological talent, and promote the coordinated development of tourism, urbanization, technology, and the ecological environment.

Eco-environmental pressure showed a significant positive correlation at a 1% significance level; this means that the increase in industrial sulfur dioxide emissions will improve the uncoordinated coupling degrees for tourism, urbanization, technology, and the ecological environment. A large number of industrial sulfur dioxide emissions will cause ecological damage to water resources and the air, and have a great impact on the health of animals and plants and human beings, even leading to death and the reduction of species diversity and affecting the balanced development of nature. At the same time, it places great constraints on the development of tourism, and thus is not conducive to the coordinated development of tourism, urbanization, technology, and the ecological environment.

The eco-environmental response shows a significant negative correlation at a 1% significance level. The improvement in the harmless treatment of municipal solid waste will reduce the uncoordinated coupling degrees for tourism, urbanization, technology, and the ecological environment. The harmless treatment of municipal solid waste refers to a remedial measure used to address people’s damage to the ecological environment. The higher the harmless treatment rate of municipal solid waste, the higher people’s awareness of the importance of protecting the ecological environment will be. Urban construction belongs to healthy development and can better realize the sustainable green development of the region.

There is a significant negative correlation between technology investment at a 1% significance level. An increase in technology expenditure will reduce the uncoordinated coupling of tourism, urbanization, technology, and the ecological environment. Technology form the primary productive force. The greater the expenditure on technology, the more it will play an irreplaceable role in promoting the development of regional productive forces, laying a solid foundation for the development of urbanization.

In order to ensure the robustness of the results, the samples were then divided into two periods (2010–2015 and 2016–2020) and the regression was carried out again. The results are shown in Table 6.

According to the results of the robustness test, there is a significant positive correlation between eco-environmental pressure and degrees of uncoordinated coupling in the two periods of time. There is a significant negative correlation between the tourism scale, economic urbanization, the eco-environmental response, technology investment and the level of uncoordinated coupling, and the result is very stable. The regression results for social urbanization from 2010 to 2015 are positive but not significant, while the regression results from 2016 to 2020 show a significant negative correlation, indicating that the research results are stable and reliable.

## 6. Discussion

Tourism, urbanization, technology and ecology are closely linked. In the rapid development of economy, the interaction of the four presents a non-benign interaction phenomenon, which belongs to the stage of regional development and has a certain inevitability. Most of the existing studies have studied the coupled and coordinated relationships among two or three of tourism, urbanization, science and technology, and ecological environment [11,14,36], but few scholars have conducted quantitative and dynamic studies on the relationships among the four as a whole. Therefore, this paper studies the uncoordinated coupling relationship of tourism–urbanization–technology–ecological environment from the reverse thinking and analyzes its spatial and temporal evolution characteristics, types of uncoordinated coupling and influencing factors.

First of all, the study shows that in recent years, with the implementation of a series of measures taken by the Chinese government to optimize industrial structure and adhere to green and sustainable development, tourism, urbanization, technology and ecological environment protection in various provinces have all developed to varying degrees [37,38,39]; this paper analyzes the comprehensive development level of tourism, urbanization, technology and ecological environment in Yunnan–Guizhou–Sichuan region, and finds that the comprehensive development level of the four are all increasing at a high speed, which verifies the existing research results.

Secondly, from the evolution of uncoordinated coupling degree, the tourism–urbanization–technology–ecological environment in Yunnan–Guizhou–Sichuan region has basically achieved sustainable and coordinated development, but the problem of uncoordinated development among cities is prominent. In a previous research, Zhang, et al. [31] found that the coupling coordination degree of economy, tourism and environment in western China has been improved; this finding is similar to the results of this study. In addition, the results of this paper show that it is necessary to implement regional cooperative governance measures to realize the coordinated development of tourism–urbanization–technology–ecological environment; this result was consistent with the new regionalism principle put forward by Ethier [40].

Thirdly, differences in eco-environmental pressure, tourism scale, economic urbanization, social urbanization, eco-environmental response, technology investment significantly affect the degree of uncoordinated coupling. Similarly, Shen, et al. [41] see technological innovation as an external driver for achieving high quality development in tourism and urbanization. In a previous study, Chenghu, et al. [42] argued that a large increase in tourists would degrade the environment and be detrimental to the coordination between tourism development, technological innovation, urbanization, and environmental quality; however, this paper holds that the increase in the number of tourists represents the improvement of the development of tourism, providing more employment opportunities for local residents, promoting the local economy, and is conducive to the harmonious development of tourism, urbanization, technology and ecological environment.

To sum up, the results not only enrich the theoretical system of coupling coordination, but also provide ideas for the formulation of Yunnan–Guizhou–Sichuan regional tourism–urbanization–technology–ecological environment coordinated development strategy. Therefore, based on the above discussion, the following policy recommendations are proposed.

**(1)** **Optimize the industrial structure and realize ecological protection**.

The ecological environment provides an important foundation for tourism, urbanization and scientific and technological development. The Yunnan–Guizhou–Sichuan region is rich in tourism resources and has a high level of eco-environmental development; however, some regions still have enterprises that cause serious pollution to the ecological environment. The government should strictly monitor and raise the threshold of environmental access. Superior departments conduct periodic random inspections of enterprises’ polluting discharge in the region, make the random inspection results public, and impose relevant punishments for the ineffective implementation of strategies by local government departments. Backward industries, such as high energy consumption and heavy pollution of the cement industry, should be accelerated to be transformed. The ecological compensation mechanism should be improved. Reasonable ecological compensation is given to restricted development areas, and severe punishment is given to areas that cross the ecological red line to realize industrial upgrades and develop an efficient, high-energy, green and diversified industrial structure.

**(2)** **Strengthen technical innovation and development**.

The source power of development comes from innovation, and scientific and technological development needs the power engine of innovation. Major opportunities brought by the new round of scientific and technological revolution should be seized. There is a long-term balanced relationship between the four subsystems (tourism–urbanization–technology–ecological environment), which promote each other while restricting each other’s development. All localities should increase investment in innovation funds, increase (instead of decreasing) investment in technology, mitigate any weaknesses, and ensure public technology activities (both basic and more cutting-edge technology research) are implemented. Preferential tax policies should be implemented for high-tech enterprises, and the recognition of such enterprises should be strengthened. The evaluation system for the transformation of technological achievements including R&D investment, income from the transformation of achievements, the proportion of transformation income to research investment, and the ratio of the number of patents to R&D investment should be improved. Further, the local government should increase the scientific research investment in universities and research institutes engaged in basic research, build a landing platform for the transformation of technological achievements of the government, enterprises and universities, and stimulate the output of major original innovation.

**(3)** **Improve tourism quality and build a tourism brand**.

At present, the problem of homogenization in tourism development is serious. Alt-hough many places are building their own tourism characteristics, more regions are eager to quickly obtain economic benefits and blindly follow the trend and build popular cities; this wastes human, physical, and financial resources. Each region should base its tour-ism on its own resource advantages, tap historical and cultural resources and build a tourism industry with regional characteristics. Additionally, skill training for personnel engaged in relevant industries should be provided, thereby improving service quality and removing homogeneous services.

**(4)** **Vigorously develop educational offerings and strengthen talent training**.

The government should increase the proportion of educational expenditure in the fiscal year. In the evaluation system of technological development, the number of college students has a high weight, which shows that talents play an important role in improving the technology. In 2020, the top three cities in terms of the number of college students are Chengdu (1,039,000 students), Kunming (698,000), and Guiyang (440,200). The cities with the fewest students are Ziyang (10,000 students), Aba (9600), and Xishuangbanna (7400). There is a great difference in the number of students in higher education school students. The scientific and technological development level in several regions that have only a small number of higher education school students is also relatively slow. Cities with lower rankings should strengthen their investment in educational resources, pay attention to the optimal allocation of educational resources between provinces and cities, recognize the professionalism of talents and improve the utlization of talented individuals. At the same time, efforts should be made to cultivate local talents and more talents with practical skills.

**(5)** **Optimize the regional layout and give full play to comparative advantages**.

According to the calculation results of Moran’s I index, the uncoordinated development between regions does not mean that differences should be wiped out at once. Each region should find its own advantage positioning by tapping its own resources. For example, Chengdu, Kunming and Lijiang, where tourism is developing well, should give full play to the comprehensive advantages of tourism, promote the development of urban catering, accommodation, shopping, shopping and transportation industries, and enhance the vitality of urban development; this will support an overall sustainable development strategy, especially in cities in the border areas. Due to the impact had by geographical location and inconvenient transportation, transportation integration would be able to stimulate the development of tourism logistics in remote areas, strengthen connections with surrounding cities, and form regional links through extensive use of online resources. At the same time, the government should take appropriate measures to promote regional coordinated development, develop suitable industries according to local conditions, and give appropriate support in terms of talents, funds, and technology.

## 7. Conclusions

The comprehensive development level of tourism, urbanization, technology, and the ecological environment in Yunnan, Guizhou and Sichuan has increased rapidly. In 2020, due to the COVID-19 pandemic, the tourism industry was the most affected, showing a straight-line downward trend. From 2010 to 2019, Yunnan province had the fastest tourism growth rate, increasing from 0.2580 in 2010 to 0.8277 in 2019, with an average annual growth rate of 94.5%. Sichuan and Guizhou province, which both developed slowly, had an average annual growth rate of 88.9% and 93.9%, respectively. Overall, the level of urbanization construction (from high to low) is as follows: Sichuan; Yunnan; Guizhou. In terms of time, the development levels of urbanization, technology and the ecological environment of the three provinces have tended to be consistent, indicating that the coordinated development among the three provinces has achieved good results.

From 2010 to 2020, the level of uncoordinated development within the three provinces of Yunnan, Guizhou and Sichuan decreased year by year; all developed from the running in uncoordinated coupling stage to a low-level uncoordinated coupling stage, indicating that the four subsystems of tourism, urbanization, technology, and the ecological environment in the three provinces of Yunnan, Guizhou and Sichuan have basically achieved a sustainable and coordinated development; however, the uneven development among the systems still exists, shown especially by the constraints placed by tourism, urbanization and technology on the development of the ecological environment.

The uncoordinated development among 46 cities at the prefecture and municipal levels in the three provinces of Yunnan, Guizhou and Sichuan is notable. The uncoordinated coupling and spatial agglomeration of various provinces and regions is relatively stable. More than 70% of cities have no significant form of agglomeration, and the main types of agglomeration are L–L and H–H. The L–L agglomeration was mostly stable in Mianyang and certain other places from 2010 to 2016, and was mainly concentrated in Guiyang from 2016 to 2020. These cities have shown rapid economic development and have close ties with surrounding cities, resulting in strong spatial relevance. The H–H agglomeration areas are mainly stable in several autonomous prefectures in Yunnan and Sichuan, and their economic development is relatively slow; their development is also closely connected with that of the surrounding cities, which have the same low level of development, thus forming low-lying gathering areas. These areas have poor scientific and technological strength, a low level of urbanization, a high proportion of agriculture, and underdeveloped transportation, all of which hinder the development of tourism to a certain extent.

The uncoordinated coupling degrees for tourism, urbanization, technology, and the ecological environment in the Yunnan–Guizhou–Sichuan region are affected by many factors. There is a significant positive correlation between eco-environmental pressure and degrees of uncoordinated coupling, and a significant negative correlation between tourism scale, economic urbanization, social urbanization, eco-environmental response, technology investment and the level of uncoordinated coupling. Other factors are not significant. If the Yunnan–Guizhou–Sichuan region wants to realize systematic coordinated regional development, it should pay attention both to these negative influencing factors and to positive influencing factors.

## Figures and Tables

**Figure 1 ijerph-19-08885-f001:**
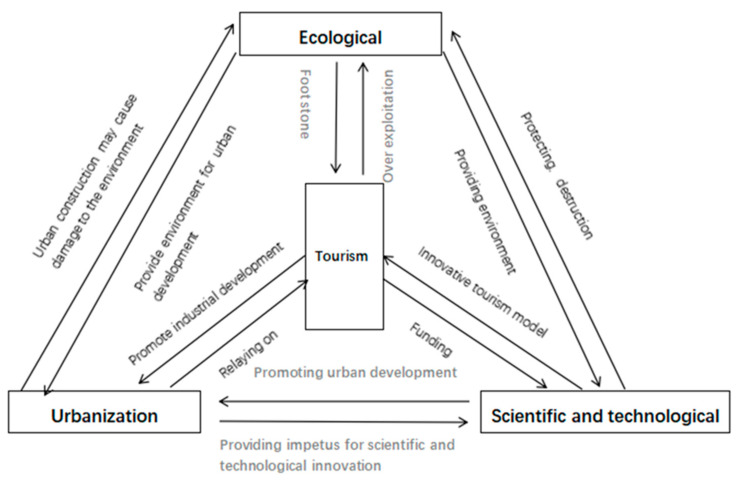
The uncoordinated coupling mechanism.

**Figure 2 ijerph-19-08885-f002:**
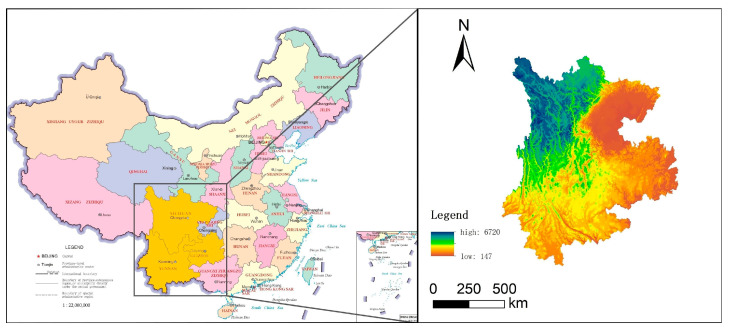
Study area.

**Figure 3 ijerph-19-08885-f003:**
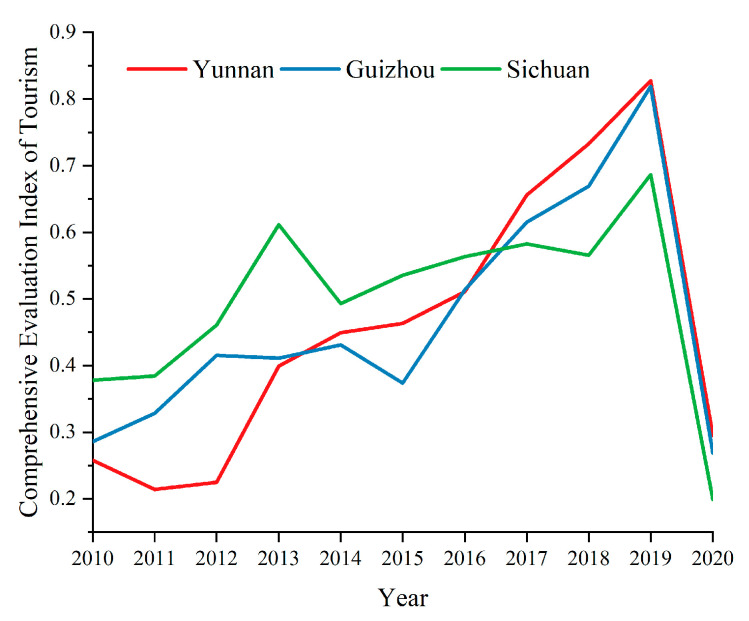
Temporal changes of the tourism development level.

**Figure 4 ijerph-19-08885-f004:**
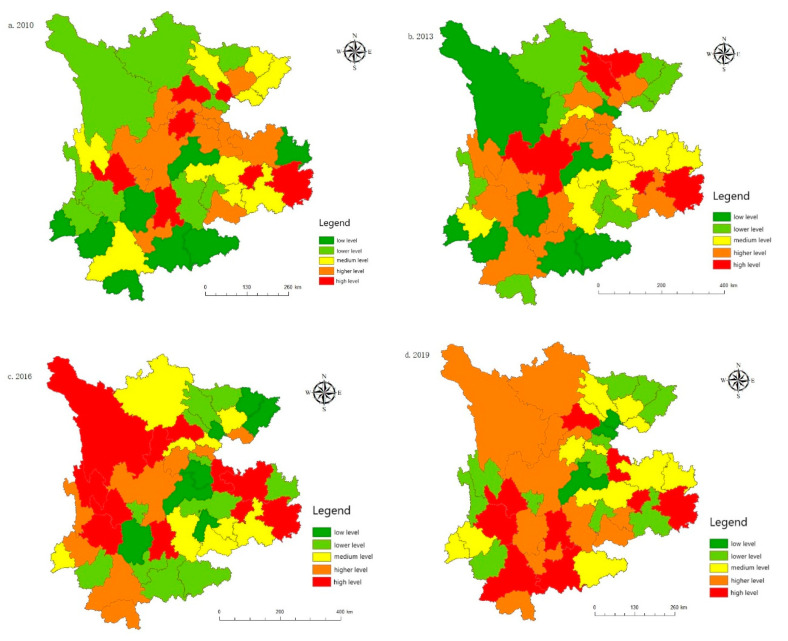
Spatial distribution of the tourism development level: (**a**) The tourism development level in 2010; (**b**) The tourism development level in 2013; (**c**) The tourism development level in 2016; (**d**) The tourism development level in 2019; (**e**) The tourism development level in 2020.

**Figure 5 ijerph-19-08885-f005:**
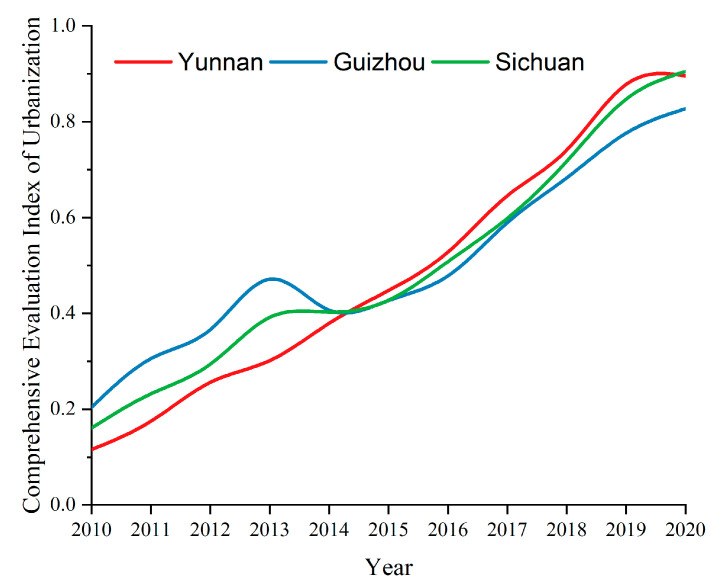
Temporal changes of the urbanization development level.

**Figure 6 ijerph-19-08885-f006:**
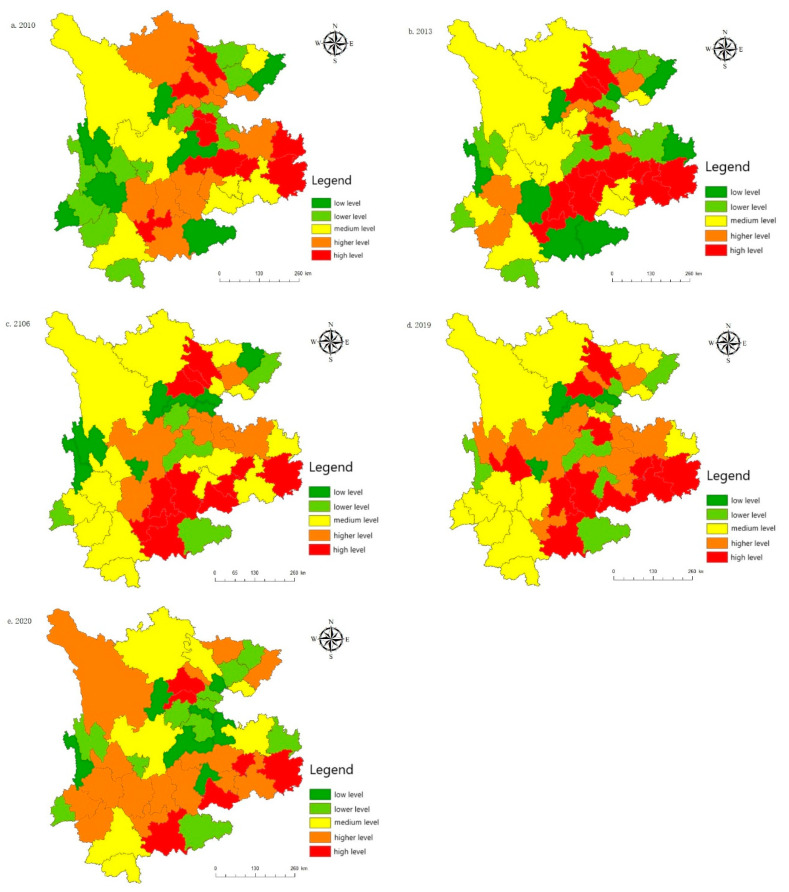
Spatial distribution of the urbanization development level: (**a**) The urbanization development level in 2010; (**b**) The urbanization development level in 2013; (**c**) The urbanization development level in 2016; (**d**) The urbanization development level in 2019; (**e**) The urbanization development level in 2020.

**Figure 7 ijerph-19-08885-f007:**
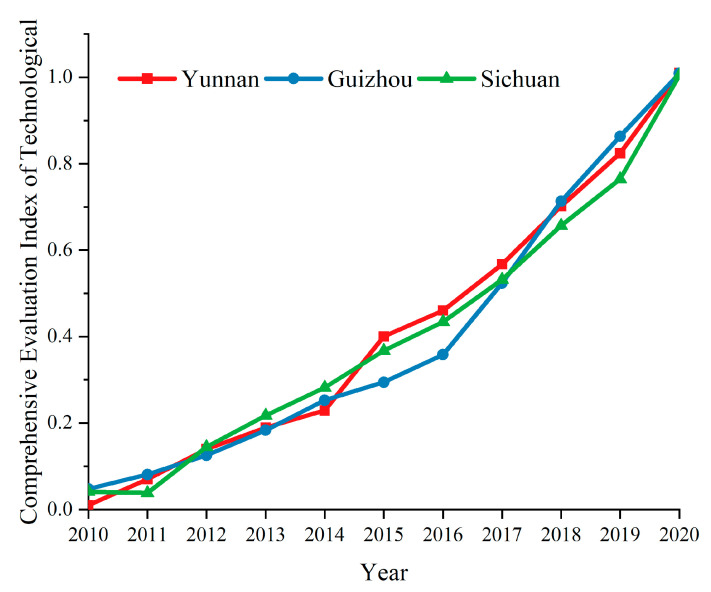
Temporal changes of the technology development level.

**Figure 8 ijerph-19-08885-f008:**
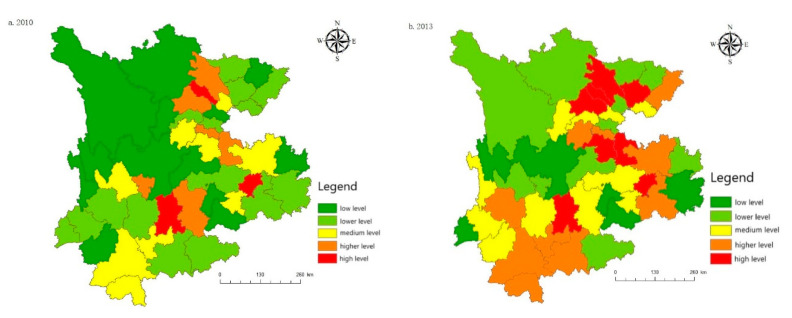
Spatial distribution of the technology development level: (**a**) The technology development level in 2010; (**b**) The technology development level in 2013; (**c**) The technology development level in 2016; (**d**) The technology development level in 2019; (**e**) The technology development level in 2020.

**Figure 9 ijerph-19-08885-f009:**
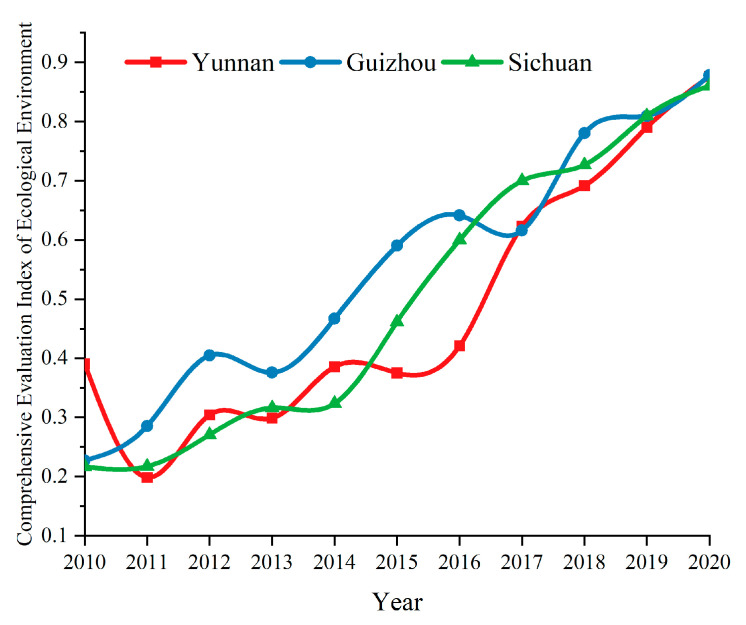
Temporal changes of the ecological environment development level.

**Figure 10 ijerph-19-08885-f010:**
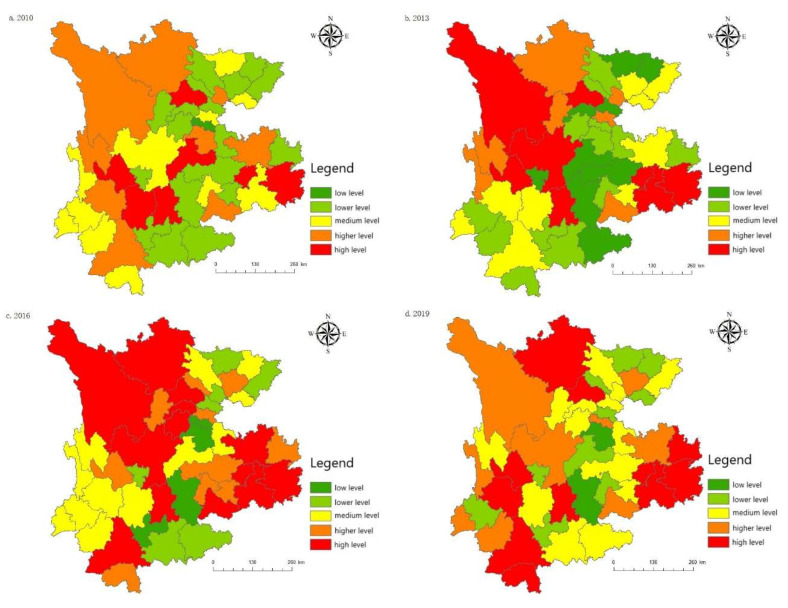
Spatial distribution of the ecological environment development level: (**a**) The ecological environment development level in 2010; (**b**) The ecological environment development level in 2013; (**c**) The ecological environment development level in 2016; (**d**) The ecological environment development level in 2019; (**e**) The ecological environment development level in 2020.

**Figure 11 ijerph-19-08885-f011:**
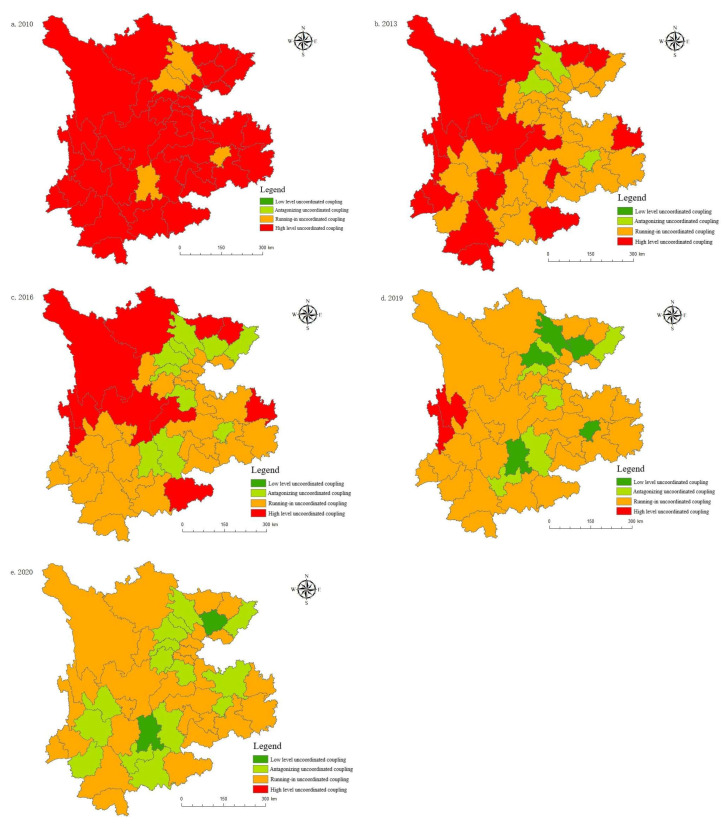
Spatial pattern evolution of the uncoordinated coupling degrees: (**a**) The uncoordinated coupling degrees in 2010; (**b**) The uncoordinated coupling degrees in 2013; (**c**) The uncoordinated coupling degrees in 2016; (**d**) The uncoordinated coupling degrees in 2019; (**e**) The uncoordinated coupling degrees in 2020.

**Figure 12 ijerph-19-08885-f012:**
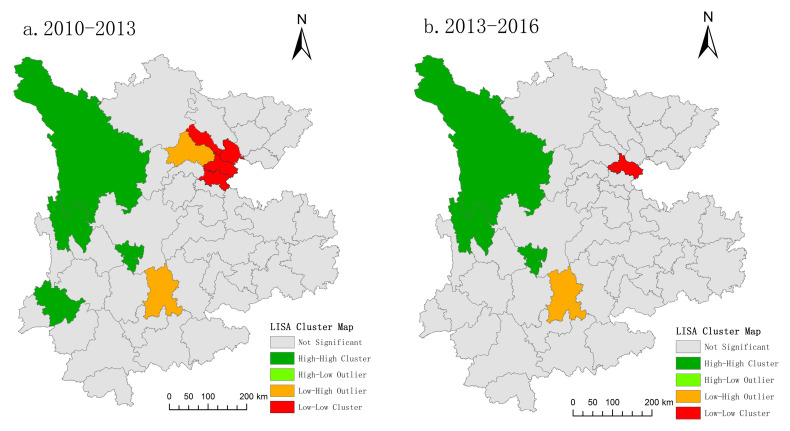
LISA agglomeration from 2010 to 2020: (**a**) LISA agglomeration from 2010 to 2013; (**b**) LISA agglomeration from 2013 to 2016; (**c**) LISA agglomeration from 2016 to 2019; (**d**) LISA agglomeration from 2019 to 2020.

**Table 1 ijerph-19-08885-t001:** Evaluation criteria for uncoordinated coupling.

Uncoordinated Coupling Degree	Uncoordinated Level	Uncoordinated Coupling Degree	Uncoordinated Level
0 < ND ≤ 0.2	Low-level uncoordinated coupling	0.5 < ND ≤ 0.8	Running in uncoordinated coupling
0.2 < ND ≤ 0.5	Antagonistic uncoordinated coupling	0.8 < ND < 1	High-level uncoordinated coupling

**Table 2 ijerph-19-08885-t002:** Comprehensive evaluation indicators for the tourism, urbanization, technology, and ecological environment system.

System	Secondary Index	Tertiary Indicators	Unit	Indicator Attribute	Yunnan	Guizhou	Sichuan
Tourism T	Tourism scale	T1 Domestic tourists	10,000 persons	+	0.2296	0.2331	0.1395
		T2 Number of inbound tourists	10,000 persons	+	0.1490	0.1132	0.1208
	Tourism benefits	T3 Domestic tourism income	100 million yuan	+	0.2640	0.2886	0.2121
		T4 Foreign exchange income from international tourism	$10,000	+	0.1233	0.1051	0.1426
	Tourism supply	T5 Number of star-rated hotels	-	+	0.1660	0.1684	0.1761
		T6 Total passenger volume	10,000 persons	+	0.0680	0.0916	0.2089
Urbanization U	Economic urbanization	U1 Regional GDP	100 million yuan	+	0.0736	0.0720	0.0729
		U2 Per capita GDP	yuan	+	0.0730	0.0677	0.0685
		U3 Disposable income of urban residents	yuan	+	0.0613	0.0649	0.0704
		U4 Proportion of secondary and tertiary industries in GDP	%	+	0.0954	0.0682	0.0745
	Population urbanization	U5 Urban registered unemployed population	Person	−	0.0319	0.0301	0.1304
		U6 Urbanization rate	%	+	0.0618	0.0776	0.0713
		U7 Urban population density	Person/square meters	+	0.0585	0.1401	0.0296
		U8 Urban population	10,000 persons	+	0.0675	0.0928	0.0733
		U9 Urban employment	10,000 persons	+	0.0775	0.0786	0.0720
	Social urbanization	U10 Number of medical institutions	-	+	0.0747	0.0436	0.0545
		U11 Health technicians per 10,000 people	Person/10,000 people	+	0.0965	0.0754	0.0695
		U12 Number of hospital beds per 10,000 people	Per bed/ten thousand people	+	0.0711	0.0653	0.0670
		U13 Gross fixed asset formation	10,000 yuan	+	0.0945	0.0735	0.0815
		U14 Real estate development and investment	10,000 yuan	+	0.0627	0.0501	0.0646
Ecological environment E	Pressure index	E1 Industrial sulfur dioxide emissions	10,000 tons	−	0.1436	0.1263	0.1139
		E2 Total discharge of industrial wastewater	10,000 tons	−	0.1400	0.0684	0.0683
		E3 Total industrial exhaust emissions	100 million standard cubic meters	−	0.0609	0.0783	0.1228
		E4 Industrial solid waste production volume	10,000 tons	−	0.0856	0.0845	0.0553
	Response indicators	E5 Industrial solid waste production volume	%	+	0.0695	0.1327	0.1191
		E6 Urban sewage treatment rate	%	+	0.0623	0.0624	0.0707
		E7 The harmless treatment rate of municipal household garbage	%	+	0.0515	0.1194	0.0908
	Status index	E8 Number of park green area per capita	Square meter	+	0.1076	0.1411	0.1066
		E9 Green coverage rate of the built-up area	%	+	0.1298	0.0794	0.1260
		E10 forest coverage	%	+	0.1490	0.1075	0.1266
Technology TE	Output index	TE1 Number of patents authorized	piece	+	0.1582	0.1402	0.1180
		TE2 Number of college students	10,000 persons	+	0.1129	0.1175	0.1042
		TE3 R&D Number of scientific and technological activity topics	-	+	0.1243	0.0986	0.1625
		TE4 Number of patent applications	piece	+	0.1440	0.1003	0.1010
	Investment index	TE5 The proportion of education expenditure in GDP	%	+	0.1308	0.1757	0.1150
		TE6 R&D personnel full-time equivalent	10,000 people/per year	+	0.1260	0.1093	0.1284
		TE7 Expenditure on technology	100 million yuan	+	0.0725	0.1180	0.1329
		TE8 R&D funds and internal expenses	10,000 yuan	+	0.1313	0.1405	0.1380

**Table 3 ijerph-19-08885-t003:** Measurement results of the uncoordinated coupling in the Yunnan–Guizhou–Sichuan region.

Time	Yunnan	Guizhou	Sichuan
2010	0.6776	0.6019	0.6093
2011	0.6114	0.5314	0.5942
2012	0.5285	0.4559	0.4807
2013	0.4639	0.4170	0.4014
2014	0.4082	0.3841	0.3942
2015	0.3519	0.3613	0.3338
2016	0.3086	0.3018	0.2772
2017	0.2113	0.2353	0.2252
2018	0.1536	0.1572	0.1854
2019	0.0893	0.0964	0.1199
2020	0.1660	0.1839	0.2076

**Table 4 ijerph-19-08885-t004:** Test results for the global Moran’s I index from 2010 to 2020.

Year	Moran’s I Index	Z Value	*p* Value
2010–2013	0.323362	3.541183	0.000398
2013–2016	0.295059	3.267267	0.001086
2016–2019	0.164881	1.925403	0.054179
2019–2020	0.140531	1.706115	0.087987

**Table 5 ijerph-19-08885-t005:** Panel regression results.

	(1)	(2)	(3)
	Mixed-Regression Tobit	Random-Effect Tobit	Fixed-Effect Tobit
	Y	Y	Y
*X* _1_	−0.206 ***	−0.062 **	−0.210 ***
	(−6.837)	(−2.152)	(−2.848)
*X* _2_	0.266	−0.185	0.071
	(0.938)	(−0.845)	(0.112)
*X* _3_	−1.152	−22.850 ***	0.685
	(−0.677)	(−4.681)	(0.252)
*X* _4_	−0.029 ***	−0.066 ***	−0.029 ***
	(−5.669)	(−11.205)	(−5.602)
*X* _5_	−0.244	−34.051 ***	−3.271
	(−0.043)	(−5.007)	(−0.468)
*X* _6_	−0.001	−0.001	−0.000 ***
	(−1.129)	(−1.577)	(−4.933)
*X* _7_	0.007 ***	0.004 ***	0.007 ***
	(5.806)	(3.105)	(5.543)
*X* _8_	−23.022 ***	−4.848	−20.490 ***
	(−4.603)	(−1.161)	(−4.836)
*X* _9_	−6.210	−42.386 ***	−5.832
	(−1.204)	(−4.358)	(−0.924)
*X* _10_	−0.004 ***	−0.007 ***	−0.005 ***
	(−4.317)	(−7.060)	(−3.259)
*X* _11_	0.000	0.000	−0.000
	(0.560)	(0.918)	(−0.264)
_cons	0.741 ***	1.057 ***	
	(14.308)	(16.227)	
Random effect or mixed effect	LR test	Chibar2 = 172.56	*p* = 0.000
Fixed effect or mixed effect	F test	F = 15.08	*p* = 0.000
Random effect or fixed effect	Hausman test	Chi2 = 237.05	*p* = 0.000
N	357.000	357.000	357.000

** *p* < 0.05, *** *p* < 0.01.

**Table 6 ijerph-19-08885-t006:** Robustness test of the uncoordinated coupling influencing factors of tourism, urbanization, technology, and the ecological environment.

	(1)	(2)
	2010–2015	2016–2020
	Y	Y
*X* _1_	−0.863 ***	−0.080 ***
	(−4.662)	(−4.282)
*X* _2_	3.995 **	−0.069
	(2.223)	(−0.393)
*X* _3_	1.955	−3.630 *
	(0.802)	(−1.887)
*X* _4_	−0.027 ***	−0.010 *
	(−3.186)	(−1.719)
*X* _5_	1.329	7.624
	(0.130)	(1.062)
*X* _6_	0.047	−0.000 ***
	(0.068)	(−3.944)
*X* _7_	0.003 **	0.006 ***
	(2.037)	(2.579)
*X* _8_	−10.388 **	−26.557 ***
	(−2.393)	(−3.558)
*X* _9_	−11.859 *	−7.023
	(−1.754)	(−1.152)
*X* _10_	−0.012 *	−0.002 ***
	(−1.717)	(−3.118)
*X* _11_	−0.000	0.000
	(−0.044)	(0.261)
N	198.000	159.000

* *p* < 0.1, ** *p* < 0.05, *** *p* < 0.01.

## Data Availability

The data used to support the findings of this study are available from the corresponding author upon request (e-mail: yangguangming@cqut.edu.cn).

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
