# Peer review of "Spatiotemporal Characteristics and Influencing Factors of Tourism–Urbanization–Technology–Ecological Environment on the Yunnan–Guizhou–Sichuan Region: An Uncoordinated Coupling Perspective"

_ijerph, 2022, doi:10.3390/ijerph19148885_

Round 1

Reviewer 1 Report

The authors examined the relationships among tourism, urbanization, science and technology, and the ecological environment across a large area of southwestern China from 2010 to 2020. The methods are sound, the conclusions are well-supported, and the manuscript is generally well-written. Most of my edits and suggestions are relatively minor.

Edits and suggestions:

Line 90: Change Meinecke to possessive: “Meinecke’s.”

Line 110: Change Poon to possessive: “Poon’s.”

Line 137: Suggest deleting “and coordination” or substituting another term to eliminate this circular definition.

Line 140: Problematic may be a better term to use than bad.  

Lines 449-453: Some editing is required here – I suggest adding “as” before “most”, capitalizing “poor,” and adding “and” prior to “the development.”

Table 4: This table seems like it may be unnecessary, as the trends and differences among areas are spelled out in the text.

Line 612: insert “be” after “usually.”

Page 28 – Policy recommendations: This section seems like the most important part of the paper, and the authors might consider using bold text for some of the sub-headings and the most important policy statements to provide emphasis, especially for government regulators and land-use planners. In addition, some of the terms are not clearly defined and few of the recommendations involve specific metrics.

Line 781: Consider providing an example of “backward industries” to help define that term.

Lines 797-800: Are there any relevant metrics to consider and strive for?

Reviewer 2 Report

Dear Authors,

Thank you for the manuscript you submitted. You analyses a very interesting topic.

Although, you need to revise most of the titles of the paper.

1. You should transform the main title in a shorter one;

2. The title 2 should be “Theoretical background”;

3. Title 3 should be presented as “Methods”;

4. Title 3.1 is not necessary;

5. Titles 2.1 and 2.2 should be much shorter;

6. Titles 3.3.1 and 3.3.2 should be eliminated;

7. The legendas of Figures 1, 3 to 11 should be shorter too;

8. The steps 1 to 11 should be transformed in i) to xi);

9. Titles 4.5 and 4.5.1 should be merged, as well as 4.6 and 4.6.1;

Reviewer 3 Report

I applaud the authors on their well written abstract and introduction, and helpful interpretation of their results. I found much of the writing clear and concise. I have a few general comments, and some specific ones, that I hope the authors will consider.

General Comments:

-Try to summarize the title of the manuscript. A long title has obvious advantages in communicating content, but if it is too long, it may be difficult to digest, inducing the reader—with little time and commitment—to move on to the next article in their literature search. 

- An application of the research must be pointed out in abstract.

- The quality of legends within all figures must be improved.  

Specific Comments:

L 34: What practical perspective?

L 243: The quality of figure 2 must be improved.

Reviewer 4 Report

The article deals with the relationship between the tourism industry, urbanisation, science and technology, and the ecological environment. The greatest merit and value of the attribution is the extensive recommendations in the last chapter. They are grouped by topic and have an applied dimension. The research was based on a difficult and complex procedure to find influences between the adopted factors. The entire procedure is described in detail with mathematical formulas and the sequence of statistical analyses. Similarly, the resulting chapter is presented, which contains as many as six steps in the procedure, the last two of which are further subdivided. This makes the article very long and the descriptive part of the method and presentation of results is disproportionately long compared to the other chapters. The reason may be that there is no separate discussion section, but it should be included in the results section. This element, however, is scarce. The reviewer suggests enriching the article with a discussion of the results, while at the same time shortening the descriptive part of the method and results somewhat (to be presented more synthetically). Currently, the text for this reason feels more methodological and like a research report, instead of presenting a higher merit (discussion of results). Furthermore, the reviewer suggests shortening the title of the article. It is very long and duplicates some of the keywords. The suggested corrections will significantly affect the perception of this text and improve its usefulness for other researchers. 
